# A Qualitative Analysis of a Caregivers’ Experience of Complementary Feeding in a Population of Native Hawaiian, Other Pacific Islander and Filipino Infants: The Timing of the Introduction of Complementary Foods, and the Role of Transgenerational Experience

**DOI:** 10.3390/nu14163268

**Published:** 2022-08-10

**Authors:** Kara Mulville, Jessie Kai, John M. Kearney, Jacqueline Ng-Osorio, Carol J. Boushey, Marie K. Fialkowski

**Affiliations:** 1School of Biological Sciences, Technological University Dublin, Grangegorman, D02 HW71 Dublin, Ireland; 2Trinity College Dublin, The University of Dublin, College Green, D02 PN40 Dublin, Ireland; 3Department of Human Nutrition, Food, and Animal Sciences, University of Hawai‘i at Mānoa, Honolulu, HI 96822, USA; 4Department of Psychiatry, John A. Burns School of Medicine, University of Hawai‘i at Mānoa, Honolulu, HI 96813, USA; 5University of Hawaii Cancer Center, University of Hawai‘i at Mānoa, Honolulu, HI 96813, USA

**Keywords:** complementary feeding, transgenerational experience, timing, Native Hawaiian, Other Pacific Islander, Filipino, infants

## Abstract

The aim of this study was to investigate caregivers’ experiences of complementary feeding (CF) among the Native Hawaiian and Other Pacific Islander (NHPI), and Filipino populations. Research focused on the timing of CF commencement, and the influence of transgenerational experience on feeding practices. The experiences and practices of those who fed human milk exclusively (HME), were compared to those who included infant formula (F&HM). Caregivers of a subset of 32 infants who were participating in a larger longitudinal study relating to CF and diet diversity, took part in voluntary in-depth interviews relating to CF practices. Interviews were recorded and transcribed. Two researchers analyzed interview transcripts. Interrater reliability and saturation were established. Institutional Review Board exemption was confirmed prior to study commencement. Interviews with 29 caregivers of infants were included in this study. Only infants of the F&HM group had an early introduction to complementary foods (<4 months of age). Caregivers reported receiving conflicting advice from healthcare professionals (HCPs) in relation to timing of the introduction of complementary foods. Nonetheless, the majority of caregivers reported following the advice of HCPs. Extended family (including grandparents) played less of a role in infant feeding, compared to previous generations. While transgenerational practices were valued and included, ultimately, the perceived health and safety of the practice for infants influenced decisions.

## 1. Introduction

Complementary foods, or any solids or liquids other than human milk or infant formula, are needed in order to meet nutritional requirements from approximately 6 months of age [1,2]. Behavioral cues of “readiness to feed” are often observed at the 4–6 month age range, correlating with when infants have physiologically developed to safely tolerate the introduction of complementary foods [3,4,5,6]. Although the World Health Organization (WHO) and the American Academy of Pediatrics (AAP) recommend exclusive breastfeeding from birth to 6 months due to its numerous benefits for both maternal and infant health [7,8,9].

Early introduction of complementary foods (EIOCF) prior to 4 months of age is associated with adverse health outcomes [10,11]. Several recent studies agree that the introduction of foods before 4 months of age (EIOCF) or after 6 months (late introduction) is associated with higher adiposity and higher obesity risk in childhood [10,11,12,13,14]. The proposed mechanisms for this relationship include early displacement of breastfeeding and its associated protective effects against obesity [15], possibly explaining why adverse effects of EIOCF are seen more strongly among infants who are formula-fed [9,10,16]. Other studies have proposed that EIOCF is a practice associated with other obesogenic early feeding practices [4,13,14,16]. The National Health and Nutrition Examination Survey (NHANES) and the National Survey of Children’s Health (NSCH) data have shown that between 2009 and 2018, the proportion of infants who had commenced CF before 4 months of age had doubled, with 32% of infants having had EIOCF between 2016–2018 [17,18]. Twice as many of these infants had been formula-fed when compared to breastfed [18].

Caregivers’ self-reported experiences of CF are scarcely reported. A review by Spyreli et al. among parents in high-income countries, found infant behavioral cues, infant size and desire to expand familial feeding roles influenced decision-making around the timing of CF commencement [19]. Demographic factors such as younger maternal age, lower levels of maternal education, and formula-feeding were also found to be associated with EIOCF [10,18,20,21]. Recent studies have found specific minority groups were less likely to be aware of the rationale behind CF guidelines [6,21]. In high-income countries, two reviews reported high levels of awareness yet limited understanding of the guidelines, with parents rejecting the guidelines as they were viewed as “too rigid” [5,19].

Native Hawaiian and Other Pacific Islanders (NHPI), and Filipinos are historically under-served minority groups in research [22]. These ethnic groups have a high prevalence of diet-related health outcomes such as obesity [23]. Non-Communicable Diseases (NCDs), including Type 2 Diabetes Mellitus (T2DM), are disparately more prevalent amongst these populations than in the general US adult population [24,25,26]. Childhood obesity is reported to affect as much as 13.6% of NHPI and Filipino children aged 2–8 years, which was the highest prevalence second only to American Indians and Alaskan Natives [23]. Oshiro et al. and Okihiro et al. investigated early rapid growth and later childhood obesity in these populations [27]. The rate of weight gain in the first 2 years of life was positively associated with BMI at the age of 5 years, and 4–5 years, respectively [16,27]. Okihiro et al. proposed that a contributing factor to these results may have been early feeding practices in these populations [16,20]. These findings support current evidence suggesting that the first 1000 days of life is a crucial nutritional window [28,29]. More research into NHPI and Filipino caregivers’ CF practices may help to guide future efforts to promote optimal early feeding practices in this population.

Culture and family were identified as factors influencing caregivers’ decision-making regarding CF such as the timing of CF and the types of foods offered [6,19,30,31]. In the Native Hawaiian culture, infants were offered fresh *poi* (steamed and mashed taro root) at 6 months [32]. Fialkowski et al. found that *poi* remains a staple in the diets of infants in Hawai‘i [33]. Similarly, taro along with coconut were traditional first foods among Other Pacific Islanders [34]. In Filipino culture, the introduction of complementary foods occurred at 6 months, with rice offered as the first food [35]. The transgenerational experience was identified as influential in caregivers’ decision-making regarding CF in several cultural groups; including the timing and types of foods [6,31]. However, these studies also found that family, in particular grandparent involvement in feeding, was associated with EIOCF [6,20,30]. In Native Hawaiian culture, *kūpuna* (grandparents), traditionally played an important role in rearing the young, and thus had a major influence on CF practices [32]. In contemporary times, the influence of *kūpuna* in CF appears to remain pertinent [36].

The aim of this secondary analysis was to explore the CF experience of caregivers of NHPI and Filipino infants, with a focus on the timing of the introduction of complementary foods and the role of transgenerational experience. We also investigated whether the chosen CF practices of those who exclusively breastfed were different from the practices of those who also included infant formula.

## 2. Materials and Methods

Prior to commencement of data collection, all participants provided written consent. The University of Hawai‘i Institutional Review Board deemed the study exempt.

This analysis was part of a larger, quantitative study exploring CF among NHPI and Filipino infants living in O‘ahu, Hawai‘i [36]. The study took place between Spring 2018 and Spring 2019. The longitudinal study investigated CF (timing and the types of food offered) among infants 3–6 months of age, and diet diversity among those 6–12 months of age. A total of 70 infants were included in the longitudinal study [36]. Inclusion criteria were: age (3–12 months), engagement in CF, and race/ethnicity (at least one of NHPI and/or Filipino). Convenience sampling, through community programs, community events as well as personal and professional networks, was used to recruit participants. The methods used for data collection in this analysis were semiqualitative.

For this study, the infant feeding method was quota-sampled from a subsample of participants from the larger study. The participants were divided into two groups based on method of infant-feeding: feeding human milk exclusively (HME) or infant formula and human milk combined (F&HM). The goal was to interview 32 participants (two groups of 16 participants each). Caregivers self-reported information on infant feeding. Caregivers were compensated with a gift card after the interview. Three foster care participants were excluded as their distinct situation and small sample size meant that analysis of their interviews would not elicit meaningful results for this sub-group. Therefore, only the interviews of caregivers (29 mothers and 2 fathers) of 29 infants were included in this study. Data replication and redundancy, indicating data saturation, occurred during the analysis of interview 8 [37]. Researchers agreed that caregivers of 29 infants would elicit sufficient data through the interview responses.

A researcher-designed survey administered through a secure online web application collected demographic details of the infants and caregivers, including infant age, sex, race/ethnicity, benefits received (Special Supplemental Nutrition Program for Women Infants and Children (WIC) [WIC]/Supplemental Nutritional Assistance Program [SNAP]), feeding since birth (HME or F&HM), household size, and cultural identity. The survey responses from caregivers of the 29 infants involved in this study were descriptively analyzed.

In-depth, semi-structured interviews provided the qualitative data used in this study. The research question guiding the qualitative data collection was: what influences decision-making on first foods of NHPI and Filipino infants? The protocol used by interviewers was adapted from research by Fialkowski et al. [35]. This protocol facilitated the interviewers’ ability to guide conversations in order to obtain relevant information and gave flexibility to acknowledge and respond to the matter at hand (such as emerging ideas). The interview involved 10 leading questions and associated follow-up questions. Participants consented for the interviews to be audio-recorded, and interviews took place at a time and private location convenient for participants.

Caregivers’ responses to a subset of 3 of the 10 leading questions related to the timing of the introduction of complementary foods and the caregivers’ transgenerational experience of CF practices were analyzed for the purpose of this study (Appendix A). Responses from caregivers who fed HME were compared to those who fed F&HM. The researchers agreed that these questions would reveal when complementary foods introduction occurred, and whether this was what caregivers were advised to do. For the purpose of this study, the CF guidelines of the WHO and AAP are used as reference for the appropriate timing of the introduction of complementary foods (from 6 months) [7,9]. The remaining 2 questions would provide insight into transgenerational CF practices among the NHPI and Filipino populations.

All interview transcripts were coded first, then key ideas and recurring themes were compared within and between groups. To ensure reliability, 4 interview transcripts were chosen at random, and 2 researchers analyzed and coded these according to a set codebook, using appropriate themes. A Cohen K of 0.72 was achieved after this initial round of analysis, signifying a level of moderate agreement [38]. Discussions occurred, and the codebook updated accordingly. Another round of analysis of 4 additional transcripts selected at random was completed by the same two 2 researchers using the updated codebook. This second round of coding resulted in a Cohen K of 0.98, which signified a strong level of agreement [38]. One researcher, using the established codebook, then analyzed the remaining 21 transcripts. The analysis of transcripts reduced the data and identified core themes and meanings [39]. Representative quotes were used to indicate transferability and qualitative trustworthiness [40]. Comparison of frequency of occurring themes between groups facilitated exploration of differences in infant feeding practices and behaviors. All participants were identified using study identification numbers with qualitative analysis occurring in Google Sheets.

Descriptive analysis of participants’ demographic data was carried out in IBM SPSS Statistics Version 27.0 (SPSS Inc.: Chicago, IL, USA). Although no statistical tests were used in analysis as this was not the intent of the study, findings are presented semi-quantitatively for the purpose of comparison of the feeding practices and experiences of the F&HM and HME groups.

## 3. Results

### 3.1. Participant Demographics

Caregivers of 29 NHPI and Filipino infants took part in this study (Table 1). In total, 13 infants were fed F&HM, and 16 were fed HME. Over half of the F&HM were fed formula since birth. Half of the caregivers in the HME group had an education attainment of an undergraduate-level degree or more, compared to under one-third of F&HM caregivers. More of the HME group were stay-at-home parents, while the F&HM group had a higher proportion of caregivers in part- or full-time employment. There was a greater variation between caregivers’ annual household income in the F&HM group. More HME caregivers lived with extended family members (including grandparents). The majority of caregivers in both groups reported Native Hawaiian as the race/ethnicity of the infant, which was similar to the breakdown of caregivers’ self-reported race/ethnicity.

### 3.2. Overview of Semi-Structured Interview Themes

The principal findings from the interviews are semi-quantitatively and thematically presented, using verbatim exemplifying quotes from the interview transcripts. A total of 25 themes and 25 subthemes were identified. Themes were grouped into two categories and their sub-categories. The first category focuses on the introduction of complementary foods, with two sub-categories: timing of the introduction of complementary foods, and principal sources of advice. The second category is transgenerational and contemporary CF practices, with three subcategories: first complementary foods, cooking and feeding roles in CF, and CF feeding behaviors and environment.

#### 3.2.1. Timing of the Introduction of Complementary Foods

A small number of caregivers in the F&HM group reported introducing foods to their infant’s diet before 4 months of age, compared to none of the caregivers in the HME group (Table 2). The majority of both groups reported commencing CF between 4–6 months of age. A higher proportion of caregivers in the F&HM group withheld CF until 6 months. One caregiver in the HME group reported late commencement of CF (>6 months).

#### 3.2.2. Principal Sources of Advice

A variation was observed in the advice, which caregivers received from healthcare professionals (HCPs; e.g., pediatricians and WIC professionals). Approximately half of such caregivers reported being advised to introduce complementary foods between 4–6 months (or earlier), while others reported being advised to withhold introduction until 6 months (Table 3). The majority of caregivers from both groups reported following the advice given by healthcare professionals (HCPs), irrespective of the timeframe advised. Conversely, fewer caregivers from both groups reported following their families’ advice. Some were advised by older family members to commence CF earlier than was recommended by HCPs. A minority of caregivers (mostly of the HME group) reported relying on their own knowledge (own research or experience raising other children), rather than other sources.

### 3.3. Transgenerational and Contemporary CF Practices

#### 3.3.1. First Complementary Foods

Most caregivers in both groups recall traditional/cultural foods being offered to infants by previous generations. Caregivers frequently reported poi, (steamed and mashed taro root) as the first complementary food offered, as was traditional for their families (Table 4). Several caregivers that fed HME, who do not self-identify as Native Hawaiian, offered poi as a first food. Some reported being encouraged by HCPs or peers, that poi was a ‘healthy’ and ‘hypoallergenic’ first food. Other caregivers also provided foods specific to their traditions and culture (Japanese, Chinese, Filipino, Tahitian and Italian). Several caregivers recalled rice cereal or premade foods being given as the first complementary foods. A higher proportion of caregivers in the F&HM group adopted the practice of providing pre-made foods for their infants. Conversely, more of the HME group described home-preparing foods.

#### 3.3.2. Roles in Cooking and Feeding Complementary Foods

Most participants recalled females (mothers and grandmothers) in their families previously undertaking the majority of such activities (Table 5). Only a small number of caregivers expressed that females (mothers) were responsible for the majority of these roles in their families today. A large number recalled males (fathers and grandfathers) having some level of responsibility in cooking and feeding in the past. Furthermore, a third of caregivers reported that, when compared to previous generations, males have a more significant role in cooking and feeding, particularly among the HME group. Grandparents in previous generations cooked and fed in recollections of nearly all of the caregivers in the F&HM group, compared with only two caregivers in the HME group. Only a small number from both groups reported that grandparents’ involvement has transcended to this generation. The same trend was seen in relation to the role of extended (non-parent) family members.

#### 3.3.3. CF Behaviors and Environment

Caregivers from both groups remembered solid food (e.g., rice cereal, poi) being added to infants’ bottles (Table 6). One caregiver explained that the belief was it helped infants to sleep for longer periods. Only one caregiver (of the F&HM group) continued this tradition. Others chose not to adopt this practice as they were advised by HCPs that it was unsafe. Pre-mastication of foods for infants was a recollection shared by a third of caregivers (equally among groups). Over half of these caregivers (mostly of the HME group) reported continuing this tradition. The main reason cited for discontinuing this practice was concerns about infant safety with several caregivers (mostly of the HME group) expressing that they were being more cautious in their decision-making relating to infant feeding, compared to previous generations. Regarding the CF environment, a third of caregivers reported feeling that poi, in particular, is increasingly more available, facilitating its inclusion in their infant’s diet. A small number of caregivers felt that the price of certain products (formula and traditional foods—including poi) has increased. Others described how poi, for them, has become more accessible thanks to programs such as WIC alleviating the burden of cost.

## 4. Discussion

This study is the first to investigate primary caregivers’ self-reported CF experiences among NHPI and Filipino populations. The findings on the timing of CF commencement agree with existing research which suggests that the practice of EIOCF is more prevalent among formula-fed infants than among infants fed human milk [18]. Although evidence regarding the relationship between EIOCF and later obesity is conflicting [10,11,14], research suggests that adverse effects of EIOCF most strongly impact formula-fed infants, likely due to lack of exposure to the anti-obesogenic properties of human milk [10,21]. Current research suggests that caregivers’ understanding of CF guidelines is associated with more appropriate timing of the introduction of complementary foods [21]. A concentrated effort is needed to ensure caregivers’ understanding of the potential health implications and safety concerns regarding EIOCF, particularly among those who feed infants formula.

The CF guidelines of the WHO and AAP recommend that CF should commence from 6 months of age [7,9]. However, half of the caregivers reported receiving advice from HCPs that CF can commence earlier than this. Other qualitative studies have reported similar findings regarding the inconsistency of CF advice among HCPs [41,42]. Mothers in one study had less trust in HCPs after advice given regarding the timing of CF commencement varied from the WHO guidelines [41]. Despite the conflicting advice, the majority of caregivers in both groups reported following the advice of HCPs regarding the timing of CF commencement. This finding was unexpected based on research among other populations [19,21]. However, in order to safeguard caregivers’ trust in the advice of HCPs, consistency and clarity of CF advice is necessary [41].

Among other traditional and cultural first complementary foods, caregivers frequently introduced poi. Poi was the most frequently offered first complementary food among infants in the larger study, of which these participants were a subsample [36]. The introduction of poi has a traditional basis. Poi is a traditional first food offered by the Native Hawaiian population [32]. Similarly, taro is a staple in the diet of many Other Pacific Islanders [33]. The WHO also encourages the inclusion of culturally appropriate foods during the CF period [7]. However, this study conveys that poi has grown in popularity across different cultural groups. This is likely because, as caregivers described, poi is readily available, and is promoted widely in the community and by HCPs. Poi is hypoallergenic due to its low protein content, easily digestible, texture-appropriate, and a source of several micronutrients (e.g., several B-vitamins, calcium, magnesium, potassium) of specific importance during the CF period [43,44]. The widespread use of poi as a first complementary food among the NHPI and Filipino populations can be considered an appropriate, if not a beneficial infant feeding practice that has transcended from previous generations. Some caregivers expressed that poi has become more accessible to them as a result of government-funded programs such as WIC. Recent findings by Campbell et al. similarly suggested that WIC/SNAP participation among NHPI and Filipino infants facilitates the inclusion of foods from ‘healthy’ food groups [45].

As was anticipated based on current research, males (fathers) were reported to have a more prominent role in cooking and feeding activities than in previous generations [46]. However, grandparents’ role in cooking and feeding was much less prevalent than is traditional for families of the NHPI population [35]. Although, the small sample size in this study is acknowledged. This observed change in feeding roles could be related to caregivers’ frequent expressions of disagreement with CF advice offered by family members, which often contradicted that of HCPs. This is in line with current research, which suggests that grandparent involvement in feeding is associated with earlier CF [20,21].

Another theme that emerged was a belief among some caregivers to be more cautious when making decisions regarding CF, compared to previous generations. This belief was also identified among caregivers in another study [6]. Likewise, some caregivers expressed distrust in pre-made infant foods and reported exclusively feeding “organic” varieties of foods, although, such beliefs are not supported by current evidence [47,48,49,50]. Similarly, most caregivers in both groups were receptive to the advice of HCPs relating to unadvised feeding practices, including pre-mastication. Pre-mastication is a transgenerational feeding practice, which holds cultural significance in the Native Hawaiian population [35]. This practice nourishes the infant’s soul [51]. However, due to concerns regarding links with communicable diseases [52], some caregivers chose not to adopt this CF practice. The discontinuation of this traditional practice based on the advice of HCPs suggest that participants’ CF choices are becoming more cautious.

The limitations of the study may be that convenience sampling may have contributed to selection bias. However, the inclusion criterion minimized recall bias, as infants included were engaged in CF at the time of the study. Demographic characteristics of participants (race/ethnicity, sex, education level) were not equally represented so there is the potential for confounding results, as such demographic factors have previously been associated with the timing of solid food introduction [10,18,20,21]. Similarly, going forward, quantitative analysis of outcomes including diet-related disease and anthropometric measures (e.g., weight, length, and other body composition indicators) would allow for more definite conclusions to be drawn from research in this area. Conversely, a strength of this study is that this is the first study to investigate caregivers’ CF experience in NHPI and Filipino populations. This investigation of under-researched racial/ethnic groups will add to the current literature surrounding NHPI and Filipino populations. Investigation of other confounding factors is warranted in future larger studies among these populations. Similarly, our findings will add to existing research regarding infant feeding methods (breastfeeding and the inclusion of formula) and CF practices. The presence of infants who were exclusively formula-fed in this sample would have been beneficial, as the practices of this group could be compared to the HME and F&HM groups. Finally, the participants may have been influenced by the interviewer, by feeling compelled to give certain responses. Open-ended and exploratory questions minimized bias by generally prohibiting simple agreement or disagreement responses. A notable strength of the study was that the interviews were conducted by researchers who self-identified as the same race/ethnicity as the interviewees. This may have put participants at ease and facilitated reassurance and comfort in sharing personal experiences.

## 5. Conclusions

Similar to other populations, EIOCF among NHPI and Filipino infants was more prevalent among those who were fed infant formula. Despite this, NHPI and Filipino caregivers were more receptive overall to the advice of HCPs relating to the timing of CF commencement than was expected based on existing research [19,21], with the advice of HCPs often overruling the advice from family. Caregivers also expressed the health and safety of their infant to be their priority during the CF period and took caution in making CF decisions. Importantly, the desire to uphold transgenerational feeding practices, such as the inclusion of poi, was evident among NHPI and Filipino caregivers. Ultimately, these findings provide a richer understanding of NHPI and Filipino caregivers’ chosen CF practices, and insight into the nutritional intake during these infants’ first 1000 days of life, which is a critical nutritional window [28]. However, in order to promote appropriate CF practices, HCPs must provide clear, consistent CF advice, that is in compliance with current CF guidelines. Further research into NHPI and Filipino caregivers’ experiences in relation to wider early feeding practices is necessary to fully assess the role of caregivers’ chosen feeding practices in the development of childhood obesity among these populations.

## Figures and Tables

**Table 1 nutrients-14-03268-t001:** Demographic characteristics relating to the caregivers and the 29 Native Hawaiian and Other Pacific Islander and Filipino infants who participated in in-depth interviews relating to complementary feeding.

Caregiver and Infant Characteristics	Total *(n)*	Formula and Human Milk Group (*n* = 13)	Human Milk Exclusively Group (*n* = 16)
**Caregiver’s Age in Years, mean (SD) ^a^**	30.7 (±5.2)	31.5 (±5.4)	29.9 (±4.9)
**Infant’s Age in Months ^b^, mean (SD)**	7.1 (±2.1)	7 (±2.5)	7.5 (±1.8)
**Age of First Complementary Food Introduction in Months, mean (SD)**	5 (±1.3)	4.7 (±1.7)	5.2 (±0.9)
**Average Household Size, mean (SD)**	4 (±1.4)	4 (±1.6)	4 (±1.3)
**Infant Race/Ethnicity, *n*(%):**			
Part Native Hawaiian or Native Hawaiian Only	18 (62)	7 (54)	11 (69)
Pacific Islander Only ^c^	1 (3)	0 (0)	1 (6)
Part-Filipino or Filipino Only	10 (35)	6 (46)	4 (25)
**Infant’s Sex, *n*(%):**			
Male	12 (41)	4 (30)	8 (50)
Female	17 (59)	9 (70)	8 (50)
**Infant Feeding Since Birth, *n*(%):**			
Formula and Human Milk Since Birth	7 (24)	7 (55)	0 (0)
Human Milk at Birth with Formula Introduced Within 3–6 Months After Birth	6 (21)	6 (45)	0 (0)
Human Milk Exclusively Since Birth	16 (55)	0 (0)	16 (100)
**Age Bracket at Which Complementary Foods Were First Introduced, *n*(%):**			
Before 3 Months	3 (10)	3 (23)	0 (0)
Between 4–5 Months	14 (49)	4 (31)	10 (63)
At 6 Months	11 (38)	6 (46)	5 (31)
After 6 Months	1 (3)	0 (0)	1 (6)
**Household Members, *n*(%):**			
Parents Only	6 (21)	3 (23)	3 (19)
Parents and Siblings Only	14 (48)	7 (54)	7 (43)
Immediate and Extended Family ^d^	9 (31)	3 (23)	6 (38)
**Caregiver’s Race/Ethnicity, *n*(%):**			
Part-Native Hawaiian or Native Hawaiian Only	13 (45)	6 (46)	7 (44)
Other Pacific Islander Only ^c^	1 (3)	0 (0)	1 (6)
Part-Filipino or Filipino Only	5 (17)	4 (31)	1 (6)
Other ^e^	10 (35)	3 (23)	7 (44)
**Estimated Annual Income of Caregiver’s Household, *n*(%):**			
<USD 35,000	5 (17)	4 (31)	1 (6)
USD 35,000–75,000	7 (24)	1 (8)	6 (38)
>USD 75,000	14 (49)	7 (53)	7 (44)
No Response	3 (10)	1 (8)	2 (12)
**Caregiver’s Highest Level of Education Attainment, *n*(%):**			
High School Diploma or Less	8 (28)	4 (31)	4 (25)
Some Undergraduate Level Education	9 (31)	5 (38)	4 (25)
Undergraduate Level Degree or Higher	12 (41)	4 (31)	8 (50)
**Caregiver’s Employment Status, *n*(%):**			
Employed (Full or Partial) ^f^	23 (80)	11 (85)	12 (75)
Unemployed ^g^	2 (6)	2 (15)	0 (0)
Stay-at-home	4 (14)	0 (0)	4 (25)
**Received Government Assistance for Food Purchase, *n*(%):**	12 (41)	7 (54)	5 (31)
Received SNAP/NAP/EBT ^h^	6 (21)	4 (31)	2 (12)
Received WIC Benefits ^i^	12 (41)	7 (54)	5 (31)

^a^ (SD) = Standard Deviation. ^b^ Age at time of interview. ^c^ Caregivers only identified themselves or their infants with Pacific Islander ethnic group(s), including Chamorro, Samoan, Tongan, Māori, Tahitian, and others not specified. ^d^ Extended family includes grandparents. ^e^ Caregivers reported identifying as Hispanic, White, Japanese and Chinese. ^f^ Employment includes full-time, part-time and self-employment. ^g^ Unemployment includes long-term and short-term (<1 year) unemployment, inability to work, fishing or farming work, and retirement. ^h^ Supplemental Nutritional Assistance Program (SNAP), Nutritional Assistance Program (NAP), Electronic Benefits Transfer (EBT). ^i^ Special Supplemental Nutrition Program for Women, Infants, and Children (WIC).

**Table 2 nutrients-14-03268-t002:** Themes and subthemes related to the Timing of the Introduction of Complementary Foods, with exemplifying quotes taken verbatim from in-depth interviews with Native Hawaiian and Other Pacific Islander, and Filipino caregivers, *n* = 29.

		Formula and Human Milk Group (*n* = 13)		Human Milk Exclusively Group *(n* = 16)	
Theme	*n* (Total)	*n*	Exemplifying Quotation	*n*	Exemplifying Quotation
Early Introduction of Complementary Foods	3	3	I think he was about a month old.	0	
			We introduced it (poi) to him when he was like 2 months, 2–3 months.		
Introduction Between 4–5 Months	14	4	At her four-month appointment, her doctor said that we can start feeding her. So we did.	10	I think he was 4ish, a little over 4 months when I gave him poi.
			Five and a half months.		He was about 5 and a half months.
Introduction at 6 Months	10	5	Yeah, rice cereal at six months.	5	I think it was about 6 months.
Late Introduction of Complementary Foods	1	0		1	I would say six-and-a-half months.
No Response	1	1		0	

**Table 3 nutrients-14-03268-t003:** Themes and subthemes related to Principal Sources of Advice for the Timing of the Introduction of Complementary Foods, with exemplifying quotes taken verbatim from in-depth interviews with Native Hawaiian and Other Pacific Islander, and Filipino caregivers, *n* = 29.

			Formula and Human Milk Group (*n* = 13)		Human Milk Exclusively Group (*n* = 16)	
Theme	Subtheme	*n* (Total)	*n*	Exemplifying Quotation	*n*	Exemplifying Quotation
Caregivers Report Not Following the Advice of Healthcare Professional	Healthcare Professional Advised Introducing Complementary Foods at 6 months	4	2	I think he was about a month old (when we introduced solids). They (the pediatrician) told me to wait till six months.	2	At four and a half months he was already showing all the signs that he was ready to eat. So, he started eating solid food. But the doctor did say if I could wait until six months, that would be better. I think all babies are different from each other. Signs are a better way of knowing than age.
	Healthcare Professional Advised Introducing Complementary Foods Before 6 months	3	1	The pediatrician wanted me to start rice cereal and stuff at 4 months. But I didn’t listen to it because I felt yeah, that’s too early. I would like her to be exclusively at least fed milk and formula or whatever for longer.	2	I think it was after six months (when we introduced foods). She (the pediatrician) was telling us by three and a half, four months that he could taste food because it was Thanksgiving and Christmas. She’s like, “He could taste some of your meals.” I’m like, “What?”
Caregivers Report Following the Advice of Healthcare Professional	Healthcare Professional Advised Introducing Complementary Foods at 6 Months	9	4	When I was at WIC (at 6 months), they told me I could start giving him solid foods. So I did. Yeah (I did what they said) but it was time. He wanted to eat solid food by then.	5	Well, the doctor recommended doing it at about six months if she looked interested. So I thought I’d try it out. She would only have maybe a spoonful here and there.
	Healthcare Professional Advised Introducing Complementary Foods Before 6 Months	11	6	And at her four-month appointment, her doctor said that we can start feeding her. So we did.	5	She told me that “if you want to start feeding him at 3 months or 4 months you can start when he starts opening his mouth and is interested.” So, with my other kids, I waited for 6 months because that was the rule, but then I started to follow his cues this time, like she advised.
Caregivers Report Not Following the Advice of Family Members		3	2	Well, actually, my mother-in-law was trying to get me to give her food after 100 days, which is just about three months. But my doctor said not yet. And so, that was the only advice for me, and I told her “No, we’re not giving her food.”	1	I mean my mom has a lot to disagree with what I feed (my baby) and how I feed (my baby).
Caregivers Report Following the Advice of Family Members		1	0		1	He was about 5 and a half months. The doctors say like 6 months, but my mom was like he can start eating. A lot of people said that they started feeding their kids earlier.
Caregivers Report Self-Knowledge Guided Decisions Regarding Complementary Food Introduction		4	1	I think with my other 3, I did (start feeding earlier) because we were actually living with my husband’s family and they are Samoan. So, they would watch my other kids, so I feel like they gave them more table foods sooner, but not with this one. This one was my small one, the one in the NICU for like 2 weeks, so I’m more careful with this one.	3	I worked at the daycare before I had kids, so I learned all about the USDA guidelines, and how much to feed, and what to start off with as baby’s first food and stuff. That’s where I learned it. Honestly if I didn’t have that, probably YouTube, I would have to do my own research. The doctor doesn’t really tell you much other than, “Oh yeah, you can start with vegetables and fruit, and make sure you watch out for allergic reactions.”
Caregiver was Not Advised about Timing of Complementary Feeding		1	1	You know I think she (her mother) just forgot- she was like ‘‘oh’’ like, she didn’t remember you know? I would tell her “Oh you know she’s going to start solids like at six months or whatever” and she’s like “Really? I don’t remember it being like that.”	0	

**Table 4 nutrients-14-03268-t004:** Themes and subthemes relating to Transgenerational and Contemporary Complementary Feeding Practices: First Complementary Foods, with exemplifying quotes taken verbatim from in-depth interviews with Native Hawaiian and Other Pacific Islander, and Filipino caregivers, *n* = 29.

			Formula and Human Milk Group (*n* = 13)		Human Milk Exclusively Group (*n* = 16)	
Theme	Subtheme	*n* (Total)	*n*	Exemplifying Quotation	*n*	Exemplifying Quotation
Traditional and Cultural Foods	Transgenerational Experience	20	9	My parents gave us poi when we were growing up, or when we were babies. And my in-laws, they had the same suggestion too, like “Try poi.” So for all of us, for my sister, my husband and his brothers, we all grew up eating poi.	11	I always remember poi. A very common food item that we grew up with that is not Hawaiian necessarily but similar to Hawaiian poi which is made from taro (kalo). But in Tahitian we have a dish called Po’e which is similar- it’s cooked bananas and it’s made into like a pudding. It’s super delicious. In my mother’s culture when needed, they would rely on coconut milk before formula. Formula was so foreign, it’s really only an American thing.
	Continuing Practice	20	9	I want him to be familiar with these things (poi) because this is where he’s going to grow up. I want him to be comfortable with it. My dad and mom maintained this strong sense about how healthy, how nutritious poi is.	11	I guess probably poi (is a tradition that has remained). Poi has been more emphasized now, even with non-Hawaiian families, just with people in general growing up in Hawaii and there’s more recognition of the good health benefits of poi, and how it’s a really good food for babies. I feel like a lot of people, and non-Hawaiians included, are using poi, especially for their babies.
	Adopting Practice of Feeding Foods Not Traditional for Self-Identified Culture	4	1	Yeah. It was the pediatrician (who recommended poi), he said poi or rice cereal is anti … what is the allergy one? And it’s healthy for them. So we decided to go with the poi first.	3	I feel like I would have never given my daughter poi if I didn’t live here. I would have never even researched it or anything. Because I’m not fond of it, but she loves it.
Pre-Made Infant Foods	Transgenerational Experience	11	3	(The food was) not even hand mashed. It was out of the jar. Yeah. And then back in the 70s, poi was a dehydrated thing or there was actually ‘baby poi’. It even said baby poi on the bottle.	8	I think when we were babies my mom used to use the jarred stuff. I guess with working moms it’s just easier that way, to just buy it.
	Adopting Practice of Including Pre-Made Foods	10	6	She (her mother) would (prepare all of the baby food)- the only reason I know it because she babysits for me sometimes and then I’ll bring like all the, you know the packaged food, and she’s like “oh my gosh I wish they had this when you guys were growing up’’.	4	I think nowadays it’s so much easier (to use pre-made baby foods)- there’s a ton of options: organic whatever you want, like non-GMO, without gluten, whatever, whatever under the sun that exists now. But in my mom’s era it didn’t, so for her, it was like you’ve got to make all of it so that way you know what’s in it. In her day, I don’t think she trusted baby food as a product.
Homemade Infant Foods	Transgenerational Experience	9	5	That’s what my sister ate and my parents told me that’s what I ate too. We ate whatever we had in the yard.	4	While I don’t have personal recollection of this, I know she firstly told me that with me, she would make her own baby foods, and puree everything, because I was the first child, so everything had to be so perfect.
	Continuing Practice of Including Homemade Foods	6	3	Tinola. So it’s fish soup. I’m going to introduce it maybe at nine months or something. Because with my sister-in-law with her children, she said that the children didn’t like any commercial food, she had to wait, and at seven months then she gave them the tinola. So I’m still going to do that, because that’s the usual.	3	I think this time around (with second child), since I’m stay at home with the second one, I’m going to try to do more of making my own stuff. My mom would want me to feed the girls healthier foods like fruits and vegetables.
	Adopting Practice of Including Homemade Foods	9	2	I think it (the food I was fed as a baby) was similar minus the fact that they didn’t make my food. My mom always tells me my favorite food was the baby banana food.	7	I honestly think my mom gave me the jarred food. With me, since I’m home more, I (make food) I buy ingredients at Sam’s Club and Costco where it’s bulk. Why get jars when you can make it?

**Table 5 nutrients-14-03268-t005:** Themes and subthemes relating to Transgenerational and Contemporary Complementary Feeding Practices: Roles in Cooking and Feeding Complementary Foods, with exemplifying quotes taken verbatim from in-depth interviews with Native Hawaiian and Other Pacific Islander, and Filipino caregivers, *n* = 29.

			Formula and Human Milk Group (*n* = 13)		Human Milk Exclusively Group (*n* = 16)	
Theme	Subtheme	*n* (Total)	*n*	Exemplifying Quotation	*n*	Exemplifying Quotation
Females Cook and Feed	Recollections of Females Cooking and Feeding	21	8	Yeah anything related to the kitchen or babies was usually a woman’s role. Which is transcending until right now.	13	Okay. So, men in my family never feed babies. I don’t ever remember my dad feeding me or my sister. Or my uncles or any of my cousins. Feeding the baby was a primarily a mommy thing.
	Females Cook and Feed Similarly to Previous Generation	3	1	So, my husband or my mom help out sometimes. Mainly it’s been me since I’m home. He (husband) rarely cooks (so I do most of the cooking). When I grew up, my mom made dinners for us.	2	Yeah, that’s what I do (in relation to moms leading the feeding role). Yeah. Only if I’m busy then (my husband) is up next.
Recollections of Cooking and Feeding Roles Being Shared Across Genders		3	1	I know my Mom and Dad both fed the babies.	2	My Dad likes to cook and would make food for my sister and I. But both my mom and dad fed us and cooked our foods. It was an even split between the two.
Males Cook and Feed	Recollections of Males Cooking and Feeding	12	5	I think my dad fed her (younger sister as a baby), he fed her rice and soup. My dad is a cook so I learned from him; My dad taught me how to cook some dishes.	7	My dad cooked for us when my mom was doing her Master’s, because we had to eat.
	Males Cook and Feed Unlike Previous Generation	9	3	(Father speaking) It was always my mom who cooked. I’m a home cook now. I want to open a restaurant, I got to teach (my son) his Italian roots.	6	I would say he is much more involved. Much more than I remember my dad being or even in his family, more than his dad was. I think that is a generational thing you know? I feel like because some of those generational boundaries have eroded a little bit, now you see this much more egalitarian approach. Both he and I are in the kitchen together.
Grandparents Cook and Feed	Recollections of Grandparents Cooking and Feeding	11	9	I think while we were babies, it was more my mom and my grandma (who fed and cooked).	2	Mom and dad, and grandparents (rotated cooking and feeding duties). My grandparents were living with us.
	Grandparents Cook and Feed	3	1	Yeah (grandparents are helpful with feeding). Besides my grandma trying to feed her too much poi.	2	I think they (grandparents) are pretty good at making sure they (my children) are only eating what they should.
Everyone (Extended Family) Feeds	Recollections of Everyone (Extended Family) Cooking and Feeding	9	6	Mm-hmm (affirmative). Man or woman, anyone feeds.	3	Yeah, and everyone helped out. We all helped cook and get the food on the table. We helped with the dishes. And we helped take care of the children.
	Everyone (Extended Family) Cooks and Feeds	2	1	They’ll (infant’s uncles) feed, yeah, they’ll like, make food for my kids or like, help me feed them.	1	My eight year-old, he kind of does a lot too for us (when it comes to feeding the baby).

**Table 6 nutrients-14-03268-t006:** Themes and subthemes relating Transgenerational and Contemporary Complementary Feeding Practices: Complementary Feeding Behaviors and Environment, with exemplifying quotes taken verbatim from in-depth interviews with Native Hawaiian and Other Pacific Islander, and Filipino caregivers, n = 29.

			Formula and Human Milk Group (*n* = 13)		Human Milk Exclusively Group (*n* = 16)	
Theme	Subtheme	*n* (Total)	*n*	Exemplifying Quotation	*n*	Exemplifying Quotation
Solid Food in Bottle	Transgenerational Experience	5	2	Both of our parents said they put rice cereal in our bottles when we were young.	3	Yes. So, they’d put a little bit of poi in the bottle and make it thicker because the thought was that feeding the baby poi will help the baby to sleep longer. That is the ‘old wives’ tale’ behind the poi.
	Continuation of Practice	1	1	(I feed my baby) banana in the bottle. My mother in law did it for everybody.	0	
Pre-Mastication	Transgenerational Experience	10	5	Pre-chewing of food before giving it to the baby. I remember my Mom doing it and I did it too for my older children. But now, I don’t do it anymore. I don’t pre-chew my baby’s food.	5	We pre-chew the food for the baby. That has always been done in our family. WIC said to try not to do it, but we do. It’s a family tradition. Of course, we don’t do it when we are sick.
	Continuation of Practice	6	1	To be honest, it should only be done if it’s safe for the baby. I don’t really think it’s the best. But it does make sure the food is swallow-able. But I do pre-chew his food- only for solid food.	5	Pre-chewing the baby’s food. Again, my mom did it, I do it, and my daughter does it to my son.
	Discontinuation of Practice	6	3	It’s gross (pre-chewing). I mean I think we grew up like that, pre-chewing.	3	We do that sometimes. (pre-chewing) If I have to, then I do, I don’t prefer it because their immune system is weaker than ours. We can be immune to something and can still pass it on to them. But it’s definitely a traditional thing.
Contemporary Belief: Increased Health Consciousness Compared to Previous Generation		11	6	My husband was raised kind of poor in the sense that there were so many of them- just eating hotdogs with spaghetti, just eating spam. So, I mean, I introduce them to you know, broccoli or different items. Yeah (Vienna sausage) is not good for you, so I’m like, “we got to eat some veggies.”	5	I feel like maybe when I was younger things were more relaxed. People didn’t really look at all the ingredients. And I think nowadays, me personally, I really look at everything that’s in what I feed (my baby). I don’t want to give her any processed sugar yet. I want to maybe develop her palate more towards eating fruits and vegetables.
						The main difference between the older people in my family and me is the information I have, and the technology. I can find information on how my baby and I can eat healthy. And the technology to make food healthier is better now—I cook or buy him fresh food.
Increased Accessibility and Availability of Poi		9	4	It is high (the cost of poi), but it doesn’t matter to me because I have food stamps, and I have WIC. So, the poi goes under the check.	5	So I think it is a lot easier to get, there’s a lot more manufacturers now. The poi that I get for her I get from the farmers market.
Increased Cost of Food		4	2	Hawai‘i is always raising prices and so for me, it’s harder to feed my children what I want them to eat. Everything is getting more expensive.	2	Yeah. For something that is grown in Hawai‘i, and is from here, it’s pretty expensive. Poi is expensive. It’s 14 dollars a bag. I was like, “Oh, my goodness.”

## Data Availability

No applicable.

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
