# Peer review of "A Qualitative Analysis of a Caregivers’ Experience of Complementary Feeding in a Population of Native Hawaiian, Other Pacific Islander and Filipino Infants: The Timing of the Introduction of Complementary Foods, and the Role of Transgenerational Experience"

_nutrients, 2022, doi:10.3390/nu14163268_

Round 1

Reviewer 1 Report

This paper aims to analyse Caregivers’ Experience of Complementary Feeding in a Population of Native Hawaiian, Other Pacific Islander and Filipino Infants. The topic is relavant. From my point of view, the introduction is well-structure but take care with abbreviations, if "Filipinos" is abbreviated with "F", use it throughout the text. The same for complementary feeding (review the text).

The term "HMO" is commonly used to abbreviate human milk oligosaccharires. Please change it to somenthing else

Table 1 is confusing, where is the sd? is the number in parentheses? please use the symbol ±. Table 2, 3, 4,5 and 6 are in another format. Please standardises the format

Reviewer 2 Report

Review of the Manuscript ID: nutrients-1836048 entitled
“A Qualitative Analysis of Caregivers’ Experience of Complementary Feeding in a Population of Native Hawaiian, Other Pacific Islander and Filipino Infants: The Timing of the Introduction of Complementary Foods, and the Role of Transgenerational Experience"  
This research focused on the timing of complementary feeding commencement, and the influence of transgenerational experience on feeding practices. The experiences and practices of those who exclusively fed human milk, were compared to those who included infant formula. The aim of this analysis was to explore the complementary feeding experience of caregivers’ of Native Hawaiian and Other Pacific Islanders and Filipino infants, with a focus on the timing of the introduction of complementary foods and the role of transgenerational experience. The differences between the chosen practices of those who exclusively breastfed and practices witch included infant formula were investigated.
I consider the results obtained at work as valuable.
I kindly ask you to clarify the following issues:
1.   Please better document the relationship between replacement nutrition and cystic fibrosis
2. Could you provide additional information on these interesting, characteristic types of complementary foods used in feeding infants in the region, e.g. nutritional values?
3. it would be good to present the summary of the results obtained more clearly, especially in the section conclusions.
4. Authors should also carefully check for grammar, punctuation, and sentence structure before submitting the revised paper. 

Reviewer 3 Report

Mulville et al. nutrients-1836048" A Qualitative Analysis of Caregivers' Experience of Complementary Feeding in a Population of Native Hawaiian, Other Pacific Islander, and Filipino Infants: The Timing of the Introduction of Complementary Foods, and the Role of Transgenerational Experience" is a valuable review paper that is the first study to investigate caregivers' CF experience in NHPI and Filipino populations. Then, the reviewer was very interested in this paper and could understand the timing of CF and the influence of transgenerational experience on feeding practices. There is one suggestion from the reviewer. This study was done entirely by interviews. In the future, the authors need to do a numerical analysis of the infant's weight, body fat, and diet-related disease. Due to this, the reviewer thinks this research will be more widespread in complementary feeding.
